# Animacy Processing in Autism: Event-Related Potentials Reflect Social Functioning Skills

**DOI:** 10.3390/brainsci13121656

**Published:** 2023-11-29

**Authors:** Eleni Peristeri, Maria Andreou, Smaranda-Nafsika Ketseridou, Ilias Machairas, Valentina Papadopoulou, Aikaterini S. Stravoravdi, Panagiotis D. Bamidis, Christos A. Frantzidis

**Affiliations:** 1Language Development Lab, Department of English Studies, Faculty of Philosophy, Aristotle University of Thessaloniki, PC 54124 Thessaloniki, Greece; eperiste@enl.auth.gr; 2Department of Speech and Language Therapy, University of Peloponnese, PC 24100 Kalamata, Greece; 3Laboratory of Medical Physics & Digital Innovation, Faculty of Health Sciences, School of Medicine, Aristotle University of Thessaloniki, PC 54124 Thessaloniki, Greece; tsmatsma7@gmail.com (S.-N.K.); elias.machairas@gmail.com (I.M.); bamidis@auth.gr (P.D.B.); CFrantzidis@lincoln.ac.uk (C.A.F.); 4Department of Psychology, Aristotle University of Thessaloniki, PC 54124 Thessaloniki, Greece; papadopoulouvg@gmail.com; 5School of Computer Science, University of Lincoln, Lincoln PC LN6 7TS, UK; kate_st@outlook.com

**Keywords:** autism, animacy, event-related potentials, social skills, picture naming

## Abstract

Though previous studies with autistic individuals have provided behavioral evidence of animacy perception difficulties, the spatio-temporal dynamics of animacy processing in autism remain underexplored. This study investigated how animacy is neurally encoded in autistic adults, and whether potential deficits in animacy processing have cascading deleterious effects on their social functioning skills. We employed a picture naming paradigm that recorded accuracy and response latencies to animate and inanimate pictures in young autistic adults and age- and IQ-matched healthy individuals, while also employing high-density EEG analysis to map the spatio-temporal dynamics of animacy processing. Participants’ social skills were also assessed through a social comprehension task. The autistic adults exhibited lower accuracy than controls on the animate pictures of the task and also exhibited altered brain responses, including larger and smaller N100 amplitudes than controls on inanimate and animate stimuli, respectively. At late stages of processing, there were shorter slow negative wave latencies for the autistic group as compared to controls for the animate trials only. The autistic individuals’ altered brain responses negatively correlated with their social difficulties. The results suggest deficits in brain responses to animacy in the autistic group, which were related to the individuals’ social functioning skills.

## 1. Introduction

Our local visual environment incorporates too much information to fully process all of it at any given moment, and so our attention is necessarily selective. This selection is partially guided via certain visual characteristics that invoke a binary distinction of world entities into animate and inanimate entitieson the basis of a process called animacy perception. Though various features are often nested in the term animacy, an entity’s ability to engage in intentional and goal-directed movements has been treated as being synonymous with the notion of animacy [1,2]. Animate creatures, also known as agents [2,3], tend to conceptually be more privileged in individuals’ mental representations over inanimate stimuli [4,5]. This privilege has been demonstrated to be driven by the fact that animate entities are uniquely identified as being hosts of a range of social competencies, including the expression of emotions, moods, and judgments, as well as intentions and goal-directedness [3,6]. In that sense, the detection of animacy on entities may underlie our ability to infer other people’s mental states, a critical component of the theory of mind [7]. Autism spectrum disorders are neurodevelopmental conditions associated with severe impairments in social functioning, such as failure to attribute social meaning to visual displays and verbal interactions [8,9], or inability to enact social knowledge to predict other people’s behaviors [10], with knock-on consequences on the individuals’ social interaction skills across the lifespan [11,12]. The interface between animacy and social cognition raises the question of the scope of possible impairment in animacy processing in autism and its potentially cascading deleterious effects on the social functioning skills of autistic individuals. In the present study, we sought to address this question by investigating: (a) animacy effects on the picture naming performance of autistic adults as compared to non-autistic (control) adults, (b) the two groups’ high-density electroencephalographic (hd-EEG) event-related potentials (ERPs) elicited via the animate and inanimate stimuli of the picture naming task, and (c) the relation between the autistic adults’ naming performance and their social functioning ability assessed via a social comprehension task.

The studies that have so far manipulated animacy in experimental designs with autistic populations converge in supporting deficits in animacy processing in autism and may be indicative of differences between autistic and neurotypical individuals in processing intentionality. Specifically, the authors of [13] assessed adults with Asperger’s syndrome on a forced-choice social belief attribution task that comprised two types of scenarios, namely those involving intentional interactions between human figures and objects (e.g., a man watering the same flower pot twice due to a woman having previously shifted the positions of two pots while the man was away), and scenarios involving mechanical interactions between objects with no intentional contingencies between the agents (e.g., a sandcastle being washed away by a sea wave) (see [14] for the full list of scenarios). The autistic adults were found to be significantly slower than their non-autistic peers when responding to the scenarios that involved intentional contingencies (thus requiring implicit social attribution to the entities involved), while no group effect was reported for the scenarios involving contingencies derived from physical causation. In another study [15], researchers used a non-verbal virtual ball-tossing game to assess high-functioning autistic adults’ social cognition and emotion regulation skills. In this game, autistic participants identified with a computerized humanoid avatar participating in ball tossing with other highly anthropomorphic avatar-like players and were asked to self-report on their social needs and emotions after they either received the ball from the other players (inclusion condition), or got completely excluded, i.e., they received the ball close to zero times (exclusion condition). According to the findings, in the exclusion context, the autistic adults seemed to be able to recognize that they were ostracized by the rest of the players during ball-tossing, however, they exhibited significantly diminished social threat and emotion regulation levels as compared to the control adult group. Both studies [14,15] provide converging evidence that the autistic adults’ ability to process intentional contingencies between animate agents is reduced as compared to their non-autistic peers, while the authors of [14] further showed that the detection of accidental contingencies between inanimate objects is preserved in the autistic individuals.

Interestingly, deficits in animacy processing have been observed in high-functioning autistic individuals with no intellectual disability, such as the population in focus in the present study, further highlighting that atypical animacy processing is not affected by the autistic individuals’ global cognitive abilities. Specifically, [16,17] have been the first studies to assess high-functioning autistic adults’ social attribution skills through their responses to interacting geometric shapes whose self-propelled movement is perceived as a proxy for animacy and agency [18,19]. In [16], the researchers found that autistic adults were less likely to ascribe thoughts and feelings to the moving geometric shapes than their non-autistic peers, and instead described the interactions between the moving shapes using spatial (instead of affective) language. Importantly, in the same study, diminished tendencies to attribute social meaning to the animated geometrical shapes were assigned to the autistic participants that exhibited high verbal skills or “passed” advanced, second-order theory-of-mind tests, further implying a pattern of impairment specific to the autistic adults’ social abilities.

The overall findings indicate that autistic adults’ ability to detect intentional contingencies resulting from human interactions or animated objects capable of self-propelled movement falls behind their non-autistic peers. However, what is still unknown is whether this disparity stems from impairments in animacy perception or deficient attention to the contingent nature of the interactions involved in the displays. Equally unknown are the neural mechanisms underlying animacy processing in autism spectrum disorders. A recent study [20] offered evidence in favor of altered brain processing of animacy in autism. The authors employed a semantic fluency task in which autistic adolescents exhibited activation across their lateral and medial frontal regions in the living word category as compared to their typically developing peers, who tended to mainly activate their lateral frontal regions. No differences in brain activation patterns were attested for the non-living word category across the two groups. In a recent EEG-based simulation study, the authors of [21] found that neural signals specific to the animacy of depicted stimuli in picture naming in healthy adults emerge at about 200 ms post-stimulus onset and are distributed across the temporal lobe regions in a spatially and temporally stable manner, but then change direction over time anteriorly with fluctuating, non-linearly changing neural signals by 800 ms. So far, the locus of possible animacy impairments in lexical processing in autism has been largely underexplored. While the spatio-temporal dynamics of real-time animacy processing have been well-documented in healthy individuals [22,23], less is known about the neural processes underlying animacy processing in autistic adults.

The current work addressed this gap by investigating how animacy is neurally encoded in autism across time through comparing young autistic adults’ hd-EEG responses to those of a non-autistic control group on a picture naming task that involved animate and inanimate pictorial stimuli. Based on previous research [13,16,24] that has shown deficits in animacy categorization and in the attribution of intentionality to animate objects in autism, we speculated that the autistic individuals may exhibit different brain activation patterns in ERP amplitudes and latencies from their non-autistic peers at early, mid, and/or late stages of animacy processing as a reflection of the different types of information encoded in animacy. According to previous studies [25,26,27], during early processing stages, the parser analyzes visual and conceptual information, i.e., it functionally and temporally dissociates animate from inanimate conceptual representations, while late processing stages rather relate to the integration of high-level social information related to animacy, such as intentionality and goal-directedness. We thus expected that the temporal dynamics of animacy in the autistic adults would differ from the controls along the loci of animacy impairment in the former group. Furthermore, we hypothesized that the autistic individuals’ potentially altered brain responses to animacy at late processing stages would be associated with their social difficulties by affecting how autistic individuals respond to various social situations. We therefore expected that the ERPs or/and behavioral responses in the picture naming task would correlate with measures of social knowledge in the autistic group. Such findings would help to understand the relation between animacy processing and social difficulties in the autism spectrum.

## 2. Materials and Methods

### 2.1. Participants

Fourteen neurotypical adults and ten verbally able autistic adults, all right-handed and Greek native speakers, were included in this study. Both autistic and non-autistic participants were students of the local Aristotle University of Thessaloniki. Individuals in the non-autistic, control group were screened on several exclusion criteria prior to participation in the study (e.g., sensory, neurological, or psychiatric problems). All participants in the autistic group had received a formal clinical diagnosis of high-functioning autism in adolescence from multidisciplinary teams (encompassing a psychiatrist, clinical psychologist, specialized educator, social worker, and speech language pathologist) at Centers for Differential Diagnosis, Diagnosis, and Support, which constitute the national organization responsible for the official diagnosis and assessment of autism and other neurodevelopmental disorders in Greece. Autistic participants were free from other known medical conditions. One author (EP) had confirmed the diagnoses on the autistic participants with the Autism Diagnostic Interview-Revised (ADI-R), which was administered at the time of diagnosis by a reliable research assessor using standard cutoffs [28]. All autistic participants met the cutoff for autism in the three subdomains of qualitative impairments, namely social interaction, communication, and repetitive and stereotyped behavior. Participants’ full-scale IQ scores were estimated using the Greek adaptation of the fourth edition of the Wechsler Adult Intelligence Scale [29]. Full-scale IQ scores for the autistic participants were between the range of 90–109, so they fell within the range of normal intellectual functioning. *t*-tests confirmed that no significant demographic differences existed between groups in either sex, age, or full-scale IQ. Participant characteristics and statistics are summarized in Table 1.

### 2.2. Experimental Tasks

#### 2.2.1. Picture Naming

Stimuli. Pictures were selected from the study published by the authors of [30] and the Multilingual Picture database [31] (see Appendix A). The picture naming task included 101 animate (animals and persons) and 131 inanimate (clothes, fruit, vegetables, vehicles, and household objects) pictures, which were matched on their lexical variables. Specifically, a Kruskal–Wallis one-way analysis of variance was applied to test for statistical differences between animate and inanimate nouns along their lexical features, including length in letters (6.8 letters for animate vs. 6.9 letters for inanimate; *χ*^2^ = 0.32; *p* = 0.572); syllables (2.9 syllables for animate vs. 2.8 syllables for inanimate; *χ*^2^ = 0.10; *p* = 0.920); (per-million) lexical frequency, which was obtained from the Institute for Language and Speech Processing (ILSP) corpus of Greek texts (17.6 for animate vs. 20.9 for inanimate; *χ*^2^ = 0.80; *p* = 0.371); imageability (273.9 for animate vs. 298.8 for inanimate; *χ*^2^ = 0.85; *p* = 0.392); and age of acquisition (229.3 for animate vs. 242.5 for inanimate; *χ*^2^ = 2.75; *p* = 0.108), the latter two of which were both obtained from [32].

Procedure. The picture naming task was implemented through PsychoPy (1st Edition, PsychoPy, London, United Kingdom, 2007) [33], and it was conducted in Greek. Participants were instructed to sit in a comfortable, armed chair in a soundproof booth and watch a computer monitor, which displayed pictures depicting objects. The monitor was placed at a 1 m distance from the participants’ eyes, and each digitized picture stimulus was automatically displayed for 2 s at the center of the screen. The trials had the following structure: (a) a fixation cross (+) centered at the screen among two successive pictorial stimuli. In order to avoid any expectation or habituation effects, (a) the display time of the fixation cross randomly varied from 300ms to 750 ms; (b) the picture was displayed for 2000 ms or until a response was received (upon the participant’s vocal response, the picture immediately disappeared from the screen); and (c) a blank space for 500 ms was employed as an inter-trial interval. The stimuli were presented in five blocks. There was a break interval among the blocks. The duration of each break varied according to the participants’ own discretion. Participants were instructed to pay attention to the visual presentation of each stimulus and to name each picture using a bare noun as quickly and accurately as possible. Experimental trials were randomly presented to each participant. Response latencies were measured using a voice key from the onset of the pictorial stimulus to the beginning of the naming response. During the experiment, participants’ vocal responses were recorded. The CMUSphinx (4th edition, Mitsubishi Electric Research Labs, Cambridge, MA, USA, 2003) [34], which is an open-source system for automatic speech recognition, was employed for the automatic recognition of response accuracy and latency (in ms) estimation. The Psychopy output files were checked offline by two of the authors (EP and VP). Eight pictures (4 animate and 4 inanimate) were selected as training items from the picture databases and were used in the beginning of the task as a familiarization phase. The picture naming task lasted ~20 min without the breaks. The experimental procedure followed in the picture naming task is visualized in Figure 1 below.

#### 2.2.2. Social Comprehension

The autistic participants were assessed on the Comprehension test of the verbal component of WAIS-IV GR [29], which measures common sense social knowledge, practical judgment in social situations, and level of social maturation, along with the extent of development of the individual’s moral conscience. The test includes 18 questions of graded difficulty that target explanations of social situations, actions, or activities that they would be expected to be familiar with (e.g., what would you do if you found a wallet on the street?). The participant’s responses are given a 0, 1, or 2 points depending on their appropriateness, while the test is terminated when the individual fails to provide a correct or a partially correct response to three consecutive questions. Then, the total raw score is converted into a standard score. The standard score ranges from 1 to 19 and classifies the performance as high (>10 standard score), normal (8–10 standard score), and below normal range (<8 standard score).

#### 2.2.3. EEG Recordings and Analysis

We employed a Nihon Kohden, high-density EEG device (Nihon -Kohden, Tokyo, Japan) with 128 active scalp electrodes. The sampling rate was set at 1000 Hz. The recordings were performed in an electrically shielded booth, which minimized both sound and light interferences. The pre-processing stage was performed through the Matlab environment “https://www.mathworks.com/products/matlab.html (assessed on 2 July 2023) and custom-based scripts written by one of the authors (CAF) in line with previous analyses [35]. The analysis initially involved a re-reference of the raw data to a common average model. Then, digital filtering was performed as follows: high-pass filtering with a cut-off frequency at 0.3 Hz; low-pass filtering with a cut-off frequency at 120 Hz; band-stop filtering for removing the power interference signal (47–53 Hz); band-stop filtering for removing the first-order harmonics of power interference (97–103 Hz); and, finally, band-stop filtering for removing the second-order harmonics of power interference (147–153 Hz).

Independent component analysis (ICA) was subsequently performed through the EEGLAB graphical user interface [36]. This resulted in 122 independent components that were visually searched by three independent neuroscientists (CAF, SK, and IM) to identify noisy electrode sites within individual participants (such as eye blinks, muscle movements, bad electrode placement, and heart rate modulation). Then, the EEG data were epoched according to the stimulus onset, and ERPs were computed. The synchronization of the EEG recordings with the stimulus onset was performed through a photodiode sensor that produced a small white square when the stimulus appeared. Each epoch consisted of 500 ms of the pre-stimulus interval and 2000 ms after the stimulus onset.

Since we employed a high-density EEG data acquisition technique, there was a large number (*n* = 122) of independent components that enabled us to model the noise components. The visual inspection of the artifactual components was performed by three independent experts (CAF, SK, and IM) and resulted in an average of 95 components (maximum value: 102; minimum value: 88). This enabled us to remove all the artifactual components and acquire high-quality data while preserving all the 232 epochs per participant. The variation in the number of the components preserved was due to the varying amount of noise per participant. However, even the lowest number of preserved components (minimum value = 88) resulted in a large number of components that were subsequently employed for the ERP analysis.

#### 2.2.4. Event-Related Potentials (ERPs)

The stimuli were classified in terms of animacy; hence, there were 101 animate and 131 inanimate picture naming trials per participant. We then estimated the average response for the stimuli belonging to each category (animate and inanimate). This resulted in a two-dimensional ERP matrix per participant and per (±animacy) category. The ERP period consisted of the pre-stimulus interval (−500…0) and 2000 ms (0…2000) after stimulus onset. This matrix consisted of 122 rows corresponding to the 122 electrode positions, and 2500 points corresponding to the 2500 sample points that created the ERP waveform. Since the sampling frequency was 1000 Hz, this meant that each sample point corresponded to 1 ms.

Following the animacy mapping study carried out by the authors of [21], which focused on the temporal and anterior areas of the brain as the loci of semantic cognition [37,38,39], the EEG analysis of the current study was performed on the frontotemporal and anteriofrontal electrode clusters on each hemisphere. Occipital areas across the two hemispheres were also added to the analyses due to their crucial role in identifying visual properties of objects, including animacy [23,40]. In sum, there were six electrode clusters, namely: (1) left frontotemporal (FFT7H, FT7, and FTT7H), (2) right frontotemporal (FFT8H, FT8, and FTT8H), (3) left anteriofrontal (AFP1, AFF1H, AFF5H, AF3, AF7, and AFF3H), (4) right anteriofrontal (AFF6H, AFP2, AFF2H, AF4, AF8, and AFF4H), (5) left occipital (O1, I1, and OI1H), and (6) right occipital (O2, I2, and OI2H).

After averaging the ERP matrices for all the autistic and control participants separately for the animate and inanimate stimuli, we produced grand-average waveforms. These waveforms are called “grand-average” since they correspond to the average waveforms of all the participants belonging to the same group (autistic vs. controls) and the same stimulus category (i.e., animate or inanimate). We mainly used grand-average waveforms due to their high signal-to-noise ratio (SNR), which further contributes to producing clear waveforms to estimate the time windows for the ERPs. We should note, however, that these graphs have been used for the estimation of the time windows of each ERP component and for visualization purposes. The actual analysis was performed on the two-dimensional ERP matrix of each participant, as previously described.

#### 2.2.5. Analysis Plan

The statistical analyses were performed using R statistical software (version 2.14.0, Center for Statistics, Copenhagen, Denmark, 2011). Prior to the statistical analysis of accuracy rates and response latencies of the autistic group and the control group across the animate and inanimate trials of the picture naming task, responses which were judged to be invalid (i.e., disfluencies/coughs/false starts that triggered the voice key), as well as latencies that were smaller than 250 ms and over two SDs from each participant’s mean, were removed from the dataset. Only response latencies on accurate naming responses were considered for the analysis. A 2 × 2 ANOVA analysis with group (autistic group/control group) as a between-subjects factor and animacy (animate/animate) as the within-subjects factor was implemented for both accuracy and response latency measures. In case an interaction was significant, independent samples *t*-tests between the two groups were run separately for the animate and the inanimate trials to unpack the interaction. Further paired *t*-tests were run within-group to detect differences in performance between animate and inanimate trials. In the social comprehension task, performance differences between the two groups were analyzed via a one-way analysis of variance (ANOVA).

For the analysis of the ERPs, we performed a series of 2 × 2 ANOVA analyses with animacy as the within-subjects factor and group as the between-subjects factor and exported mean area amplitudes and latencies to peak within each time window as the dependent measures. Analyses were run separately for each hemisphere (left/right) and region of interest, i.e., anteriofrontal, frontotemporal, and occipital. In case of significant interactions, independent samples *t*-tests between the two groups were run separately for the animate and the inanimate trials. For reasons of brevity, we only reported on the effects and interactions that reached significance (*p* < 0.05). All the mean amplitudes and latencies (SD) for each group and each electrode cluster can be found in Appendix A, while ERP values for each autistic participant in the components exhibiting statistically significant group by animacy interactions are reported in Appendix A. Grand-average waveforms were obtained for each one of the six regions of interest, i.e., the left/right anteriofrontal, frontotemporal, and occipital areas.

Finally, given our hypothesis that the behavioral and ERPs elicited from the picture naming task may correlate with social knowledge, we ran bivariate Pearson correlation analyses between: (a) accuracy and response latencies in the animate and inanimate pictures of the naming task, and (b) EEG-exported mean area amplitudes and latencies across the various components, and the groups’ scores in the social comprehension task. For reasons of brevity, only the ERPs that significantly correlated with social comprehension performance scores in each group were reported in the Section 3. Correlation coefficients and *p*-values across all ERPs can be found in Appendix A.

## 3. Results

### 3.1. Behavioral Results

Picture naming. Removal of outliers, i.e., invalid naming responses and latencies, yielded 3.1% and 2.9% for the autistic group and the control group, respectively. Mean accuracy scores and response latencies per group are presented in Table 2.

Regarding naming accuracy, the 2 × 2 ANOVA analysis yielded no significant effects for either group (F(1, 22) = 1.773; *p* = 0.201; η^2^ = 0.08) or animacy (F(1, 22) = 1.686; *p* = 0.212; η^2^ = 0.01). However, there was a significant two-way interaction between group and animacy (F(1, 22) = 23.144; *p* < 0.001; η^2^ = 0.06). For the animate trials, we found that the control group scored higher than the autistic group in the animate trials (t(22) = 2.538; *p* = 0.021). There was no significant group effect in the inanimate trials (t(22) = 0.206; *p* = 0.839). Further paired *t*-tests within each group showed that the control adults were more accurate in the animate trials compared to the inanimate trials (t(13) = 3.143; *p* = 0.008). On the other hand, the autistic participants were less accurate in the animate trials compared to the inanimate trials (t(9) = 5.661; *p* = 0.005). Boxplots for each group’s accuracy scores in the animate and inanimate trials are shown in Figure 2 and Figure 3, respectively.

For the response latency measure, the 2 × 2 ANOVA analysis yielded no significant effect for group (F(1, 22) = 1.762; *p* = 0.202; η^2^ = 0.08). There was a significant effect found for animacy (F(1, 22) = 13.421; *p* = 0.002; η^2^ = 0.06), which stemmed from the fact that animate pictures were responded to slower than inanimate pictures. The two-way interaction between group and animacy was marginally significant (F(1, 22) = 4.142; *p* = 0.058; η^2^ = 0.02). Paired *t*-tests within each group showed that response latencies across animate and inanimate trials were not significantly different in either controls (t(13) = 1.853; *p* = 0.087) or in the autistic participants (t(9) = 3.430; *p* = 0.07).

Social comprehension task. The autistic group scored below the normal range (mean: 5.6 (SD: 1.2)) and lower than the controls (mean: 11.1 (SD: 0.9)), F(1, 22) = 6.114, *p* = 0.018, and η^2^ = 0.07.

### 3.2. ERP Results

Figure 4 illustrates the grand-average ERPs for the six regions of interest, i.e., the left/right anteriofrontal, frontotemporal, and occipital areas. In case of statistical significance (*p* < 0.05), corresponding ERP intervals have been highlighted in the figure via black rectangles. Also, Table 3 visualizes the statistically significant results across the ERP components.

In the left frontotemporal area, the analysis of the P100 latencies showed a significant animacy effect, F(1, 22) = 5.611, *p* = 0.03, and η^2^ = 0.03, which was due to the fact that latencies on the inanimate trials (mean: 123.6) were slower than on the animate trials (mean: 115.2). Also, in the slow negative wave (SNW) latencies, the ANOVA yielded a significant two-way interaction between animacy and group, F(1, 22) = 4.145, *p* = 0.048, and η^2^ = 0.03. The independent samples *t*-tests showed that the control group had longer latencies (mean: 1666.5 (SD: 215.5)) than the autistic group (mean: 1435.8 (SD: 262.5)) on the animate trials (t(22) = 2.604; *p* = 0.022). On the other hand, there was no significant difference between the two groups’ latencies on the inanimate trials (mean: 1588.8 (SD: 295.6) for controls ≈ mean: 1552.6 (SD: 293.9) for autistic) (t(22) = 0.235; *p* = 0.817).

On the right frontotemporal area, the analysis of the P300 amplitude showed a significant animacy effect, F(1, 22) = 3.981, *p* = 0.05, and η^2^ = 0.09, which was due to the fact that the amplitudes on the animate trials (mean: 1.2 (SD 1.2)) were larger than the inanimate trials (mean: 0.2 (SD: 1.1)). There was also a significant group effect on the SNW component, F(1, 22) = 11.559, *p* = 0.003, and η^2^ = 0.33, which stemmed from the fact that the latencies of the control group (mean: 831.5 (SD: 163.1)) were slower than for the autistic group (mean: 507.0 (SD: 162.7)).

On the left anteriofrontal region, there was a significant two-way interaction between animacy and group, F(1, 22) = 4.070, *p* = 0.048, and η^2^ = 0.09, in the N100 amplitude. For the animate trials, the control group’s N100 amplitude (mean: −1.3 (SD: 0.6)) was larger than that of the autistic group (mean: −0.6 (SD: 0.4)) (t(22) = 2.303; *p* = 0.028), while the reverse was found for the inanimate trials, i.e., the control group’s N100 amplitude (mean: −1.0 (SD: 0.5)) was smaller than that of the autistic group (mean: −1.9 (SD: 0.9)) (t(22) = 2.370; *p* = 0.029).

On the left occipital region, there was a significant animacy effect on P200, F(1, 22) = 4.873, *p* = 0.041, and η^2^ = 0.02, and the slow positive wave (SPW) amplitudes, F(1, 22) = 13.909, *p* = 0.002, and η^2^ = 0.02, both being due to the fact that the amplitudes on the animate trials were larger than on the inanimate trials (for P200: animate (mean: 2.3 (SD: 2.2)) > inanimate (mean: 1.6 (SD: 2.3)); for SPW: animate (mean: 4.9 (SD: 3.3)) > inanimate (mean: 3.5 (SD: 3.7))). There was also a significant two-way interaction between animacy and group, F(1, 22) = 7.156, *p* = 0.016, and η^2^ = 0.02, on P300 latencies. The independent sample *t*-tests showed that the on the inanimate trials the autistic group had shorter latencies (mean: 308.6 (SD: 24.4); t(22) = 2.604; *p* = 0.022) than the controls (mean: 347.9 (SD 42.8), t(22) = 2.022; *p* = 0.042). On the other, the two groups did not differ on their latencies in the animate trials (mean: 324.5 (SD 45.2) for controls ≈ mean: 354.4 (SD: 24.1) for the autistic group; t(22) = 1.336; *p* = 0.199.

Finally, on the right occipital area, there was a significant animacy effect on the SNW amplitude, F(1, 22) = 8.900, *p* = 0.008, and η^2^ = 0.07, which was due to the fact that the amplitude on the animate trials (mean: −4.4 (SD: 3.5)) was larger than on the inanimate trials (mean: −2.8 (SD: 1.9)).

### 3.3. Correlations

The correlation analyses yielded no significant results for the control group for the relation between performance in the social comprehension task and the behavioral measures of the picture naming task; *r*(14) = −0.162, *p* = 0.581 for accuracy in animate trials; *r*(14) = 0.376, *p* = 0.185 for accuracy in inanimate trials; *r*(14) = 0.032, *p* = 0.914 for response latencies in animate trials; and *r*(14) = 0.359, *p* = 0.208 for response latencies in inanimate trials. Similarly, there was no significant correlation between controls’ performance in the social comprehension task and the ERPs (see Appendix A). The correlation analyses between the autistic individuals’ social comprehension scores and ERP measures identified a significant negative correlation between the social comprehension scores and the left frontotemporal slow positive wave component amplitudes (*r*(9) = −0.884; *p* < 0.001) on the inanimate trials (see Figure 5) (see Appendix A for the correlation output from social comprehension scores and ERPs across all electrode clusters).We have also observed a significant negative relationship between the autistic individuals’ response latencies to the animate trials of the picture naming task and their scores in the social comprehension test (*r*(9) = −0.802; *p* = 0.005) (see Figure 6). The relation between the autistic individuals’ social comprehension scores and the rest of the behavioral measurements of the picture naming task were not found to be significant; specifically, *r*(9) = 0.134 and *p* = 0.711 for the relation between social comprehension scores and accuracy in animate trials; *r*(9) = 0.549 and *p* = 0.100 for the relation between social comprehension scores and accuracy in inanimate trials; and *r*(9) = −0.465 and *p* = 0.176 for the relation between social comprehension scores and response latencies in inanimate trials.

## 4. Discussion

The present study is the first to combine neurophysiological and behavioral metrics to investigate how autistic adults process animacy in a picture naming task. The main novelty of this study is the measurement of both spatio-temporal neural and behavioral differences of animacy processing between young autistic adults and their neurotypical peers through fusing the individuals’ behavioral performance, i.e., accuracy and response latencies on animate and inanimate pictures, with temporal ERPs in both hemispheres. In addition, the current study provides novel evidence of relations between animacy processing and social comprehension difficulties related to autism.

Our results provide evidence of animacy processing deficits in autism at both the behavioral and neural levels. In particular, the behavioral analysis showed that the autistic individuals had lower accuracy scores than the controls in the animate pictures of the task. The subsequent ERP analysis showed that the autistic group exhibited altered neural responses over the left anteriofrontal electrodes during the early stages of processing, whereby the autistic group exhibited smaller and larger N100 amplitudes than the control group on the animate and inanimate trials, respectively; also, in the left occipital P300 component (mid processing stage), the autistic adults showed shorter latencies than their controls on the inanimate trials of the picture naming task. Differences between the two groups were also reported at late stages of animacy processing, since in the left frontotemporal SNW component, the autistic adults exhibited shorter latencies than the controls on the animate trials. In addition, we found that the autistic group’s response latencies on the inanimate pictures of the picture naming task, as well as amplitudes within the SPW component over the left frontotemporal area, showed negative correlations with the autistic individuals’ social comprehension skills, as measured through a standardized social comprehension task. The overall results suggest deficits in the autistic adults’ processing of animacy at early, mid, and late stages of processing that were primarily reflected in altered neural activity in the left anterior and occipital areas between 100–300 ms and in the left frontotemporal area between 1000–1600 ms after picture stimulus onset. Finally, the correlation analyses have highlighted a functional relationship between animacy processing impairments and social functioning difficulties related to autism.

More specifically, the behavioral data from picture naming showed that the autistic adults provided considerably fewer correct responses to the animate trials of the task as compared to the control group. It is noteworthy that the autistic adults’ erroneous responses on the animate trials of the picture naming task mainly consisted of ‘no responses’, as well as non-target inanimate nouns, which were, however, semantically related to the animate target noun (e.g., ‘desert’ instead of ‘camel’, or ‘antenna’ instead of ‘ant’), further suggesting that the autistic adults’ naming ability was affected by semantically related inanimate concepts that competitively interfered with the autistic individuals’ subsequent naming of objects in the animate category. In addition to accuracy, the autistic adults were faster in naming inanimate pictures (0.98 ms) over animate pictures (1.07 ms); however, this tendency failed to reach statistical significance. Since animate and inanimate nouns in the task were matched on a range of lexical parameters, including lexical frequency, length, imageability, and age of acquisition, our results point to processing difficulties inherent to the pictures’ animacy feature. This finding is consistent with past work in autism [13,16] that claims that animate entities are perceived less efficiently than inanimate entities by individuals on the spectrum, which may be due to some fundamentally different characteristic of autistic individuals’ perception of agents as hosts of intention and deliberation in action planning [41]. For example, difficulty with perceiving animate objects has been reported by the authors of [24]’s object identification paradigm; autistic adults in [24] required more frames to identify visually impoverished animate objects than controls, while the two groups did not differ in their ability to decipher blurred inanimate objects. The strength of the animate vs. inanimate object discrepancy in autism has also been attested in child populations on the spectrum; for example, a sorting-by-preference study [42], which assessed autistic children’s preferences towards animate and inanimate entities, showed that the autistic children tended to prefer pictures depicting inanimate (vs. animate) entities to a considerably greater extent than their verbally matched neurotypical peers, as well as children with Down’s syndrome, further implying that the animacy perception deficit may be specific to autism independently of the individuals’ language and intellectual functioning skills.

The current study further supplemented autistic adults’ difficulty with the animacy domain at the behavioral level with dynamic neurocognitive evidence of the time course of naming animate and inanimate objects through hd-EEG/ERPs recordings. According to the findings, the autistic group exhibited disrupted brain responses at various temporal stages (early, mid, and late) of processing during picture naming. Specifically, on the left anteriofrontal N100 component, which represents the visual processing of stimuli [43,44], the two groups’ brain signals showed a mirroring image pattern, since the autistic adults exhibited reduced amplitudes as compared to the controls in the animate trials, but decreased amplitudes with respect to their control peers on the animate trials. Also, the autistic group showed longer latencies than the control group on the inanimate trials on the left occipital P300 component (mid processing stage), which has been associated with the categorization of stimuli in terms of their biological gender determined via animacy [45,46]. Both findings suggest that processing animate pictures may have required more effortful attentional and semantic integration processes for the autistic adults as compared to their non-autistic peers, while inanimate stimuli were less costly for the autistic individuals.

At the late stages of processing, the autistic group exhibited shorter (than the controls) left frontotemporal SNW latencies on the trials only depicting animate objects. In social neuroscience research, the late SNW component, which emerges in the 1000–1600 ms interval, has been related to the time course of neural signatures of mentalizing processes and, more specifically, to the identification of intentions and mental representations of agents involved in goal-directed behavior [47]. Studies have, for instance, reported a frontal SNW associated with the integration of perceptual visual cues (e.g., eye gaze) relevant to the mental states of protagonists in order to explain a social interaction scenario [48,49,50]. In the animate trials of the current picture naming task, we found that the autistic adults showed shorter latencies than the controls on the SNW component, which may indicate deficits in accessing high-level socio-cognitive information related to animacy. If the SNW component indeed reflects the integration of perceptual cues, including animacy, for the purpose of mental state attribution to possible agents, then autistic adults’ shorter latencies as compared to controls may provide a signature of deficient animacy processing in autism. Indeed, when animacy is not adequately signaled in the brain, it might become difficult for an individual to determine communicatively intentional, goal-directed social interactions [9].

The overall spatio-temporal dynamics of animacy processing in the autistic group adds to the existing evidence that animacy is not a monolithic construct [4,5], and that the information encoded in animacy may be impaired in the autistic group along the core animacy dimensions. Altered early-to-mid (100–300 ms) brain signals in the autistic adult group may have stemmed from difficulties with orienting attentional resources to visual traits of the depicted entities and with integrating low-level semantic features. On the other hand, the same group’s altered neuronal activity at the late stages of processing (1000–1600 ms) may reflect difficulties with the integration of high-level socio-cognitive information related to the intentionality cues encoded in animate objects.

Finally, the correlations of the current study offer support in favor of a link between autistic adults’ animacy processing ability and their social functioning skills. More specifically, SPW amplitudes over the left frontotemporal area on the inanimate trials of the picture naming task and response latencies on the animate trials were found to be negatively correlated with the autistic adults’ social comprehension scores. Slow positive potentials over the frontotemporal areas have been reported to be related to conceptual loads [51,52]; as such, the SPW component in the current study may index the conceptual load associated with the perception of animacy on objects. In the current study, the autistic adults exhibited relative ease in naming inanimate objects over animate objects; their response accuracy on the inanimate trials was higher than for animate objects (88.6% > 84.1%), while their naming latencies for inanimate trials were faster (though not at a significance level) than that obtained for the controls (0.98 ms < 1.07 ms). Also, the autistic group’s attentional resources allocated to inanimate entities were considerably greater than for the controls, as reflected in the analyses of the N100 amplitudes over the left anteriofrontal regions. Crucially, in the correlation analyses, social difficulties for the autistic group, as manifested in the social comprehension task, showed an inverse relationship with the strength of the SPW component on the inanimate trials; in other words, autistic individuals that performed low on the social scenarios of the social comprehension task tended to show higher amplitudes in the SPW component signaled via inanimate picture processing than their autistic peers with higher social comprehension scores. Also, low performers on the social comprehension task seemed to be slower in naming animate objects than their autistic peers that achieved higher accuracy scores in the social comprehension task. While further experimental research is needed to explore the causal directionality within the observed relations, we speculate that autistic individuals’ increased attention to inanimate (over animate) entities [24,42] may affect their ability to adequately attribute mental states to other individuals, which may, in turn, result in social comprehension difficulties. Interestingly, several studies have highlighted that a decreased ability to identify intentional contingencies between agents in autism underlies crucial social and mentalizing deficits in this population [53,54,55,56,57]. In the same respect, compromised processing of animacy as a proxy for intentionality may serve as a factor that contributes to social functioning deficits on the autism spectrum.

Although the results of the current study provide relatively straightforward evidence for animacy deficits and their relation to aspects of social cognition in autism, our findings should be replicated by follow-up studies given the following limitations. First, since the current study included a small sample size of participants, picture naming tasks or paradigms manipulating animacy should be replicated in future studies with larger numbers of participants to ensure that the results are robust. Based on the well-acknowledged heterogeneity in autism and the large individual differences noticed among autistic individuals across studies [58,59], improved sample sizes are required to enable broader conclusions. Furthermore, it is possible that poor animacy processing in autism may be modulated by other factors to those explored here, for example, reduced inhibitory skills, which may have prevented the autistic individuals from disengaging their excessive attention to inanimate (over animate) pictures in the picture naming task. Previous research has shown that autistic individuals, both children and adults, suffer from profound executive function difficulties with inhibition and attention shifting [60,61]. Thus, it might be of interest in future work to examine inhibitory and attentional skills alongside the processing of animacy so as to uncover other potential sources of difficulty with the animacy domain in autistic individuals. Finally, despite the employment of a hd-EEG analysis at a sensor level, neural correlates of animacy in autism could be further highlighted through cortical connectivity analyses, especially in view of the under-connectivity theories of autism that have given way to more nuanced characterizations of the neural basis of social difficulties in the disorder [62,63]. Cortical connectivity analyses, which overcome the intrinsic spatial limitations of EEGs, are currently in progress to further explore the neural underpinning of animacy in the autistic individuals.

We should further note that the current study employed a hd-EEG system with 128 electrodes. Until recently, sensor ERP studies employed a much smaller number of electrodes (e.g., 19 electrode montage) placed according to the 10–20 international system [64]. The analyses in past research tended to focus on ERP waveforms of either single electrodes [64] or electrode clusters [65]. Although such designs appear to be a feasible approach in neurophysiological research due to their relatively low cost and low effort, they only include the core electrodes defined by the 10–20 international system. For the purposes of the current study, there has been evidence from previous neurophysiological studies [21,22] that crucial regions of interest involved in animacy processing were not covered with the specific electrodes. This was particularly relevant for the posterior regions, whereby only the O1 and O2 electrodes would be employed for the left and right occipital clusters. The current experimental design allowed us to expand previous research [21,22] and enhance the resolution of EEGs by including additional electrode regions, such as I1 and OI1h for the left cluster and I2 and OI2h for the right cluster. Furthermore, although the analyses in the current study focused on identifying differences in temporal ERPs (amplitude and/or latency) between autistic adults and healthy controls along the animacy dimension, we tried to establish a high sampling rate that would allow our findings to extend to the high gamma range to support future cortical functional connectivity analyses. More specifically, most of the frequency content in the ERP waveforms was lower than the beta band (20 Hz), which facilitated the employment of low sampling rates up to 256 Hz. However, this may be a barrier to cortical functional connectivity analyses based on high-frequency oscillations (up to 100 Hz) that frequently occur in pathological conditions, including autism [66]. Therefore, along with employing a hd-EEG setting, we established a high sampling rate that extended to the high gamma range. To avoid noisy sources in high frequencies, we applied band-stop filters that removed main powerline interference centered at 50 Hz and its harmonics at 100 and 150 Hz [67].

## 5. Conclusions

The present study provides neurocognitive evidence supporting a deficit in integrating both low-level perceptual and high-level socio-cognitive information cued via animacy in a group of young autistic adults. The study is novel in that it suggests a computational framework that fuses atypical behavioral performance and disrupted spatio-temporal brain responses to animate and inanimate stimuli in a picture naming task for the autistic individuals alone. Finally, this study revealed that animacy deficits negatively affected social comprehension skills in the autistic group. The overall findings constitute the first demonstration of neural signatures of the dynamic time course of animacy integration in autism during a picture naming task, while also showing that disruptions in animacy processing may serve as a marker of social comprehension impairments that relate to the autism spectrum disorder.

## Figures and Tables

**Figure 1 brainsci-13-01656-f001:**
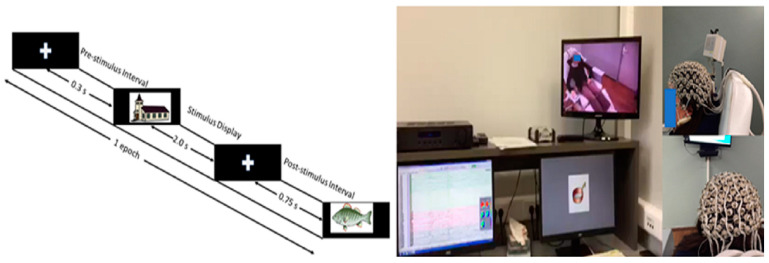
The picture naming task procedure.

**Figure 2 brainsci-13-01656-f002:**
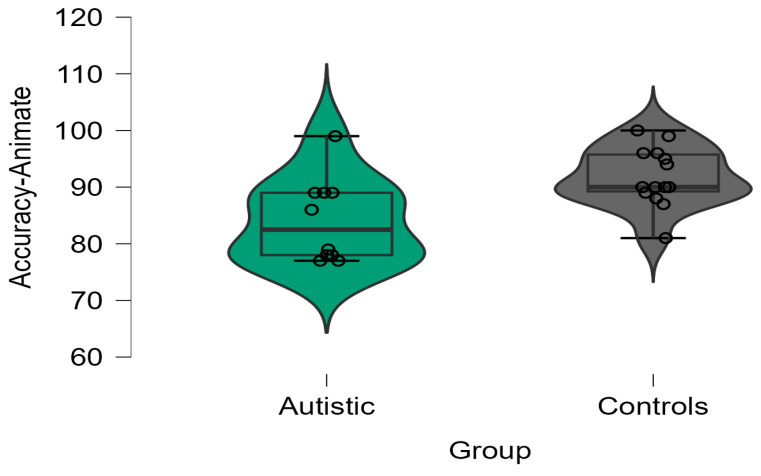
Accuracy (%) in animate trials in the autistic and control groups.

**Figure 3 brainsci-13-01656-f003:**
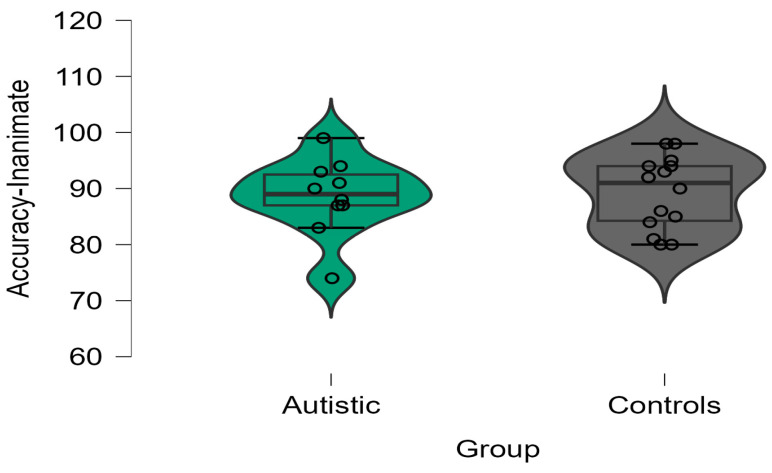
Accuracy (%) in inanimate trials in the autistic and control groups.

**Figure 4 brainsci-13-01656-f004:**
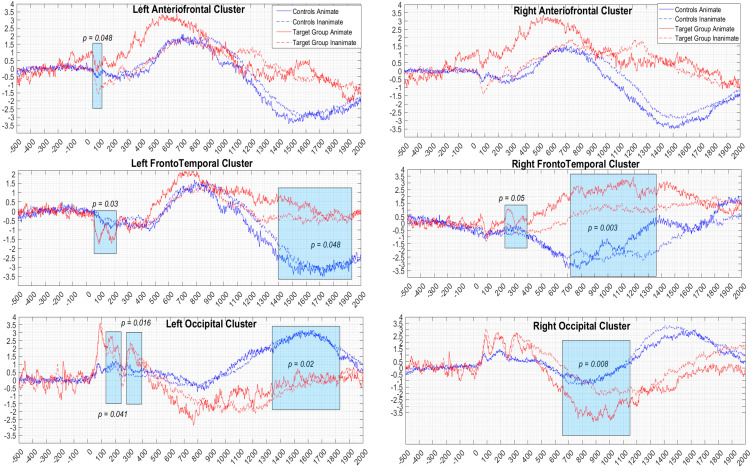
Grand-average ERPs from the picture naming task. The blue and red lines represent the control’s and the target/autistic group’s ERP responses across the animate (solid line) stimuli and the inanimate (dashed line) stimuli.

**Figure 5 brainsci-13-01656-f005:**
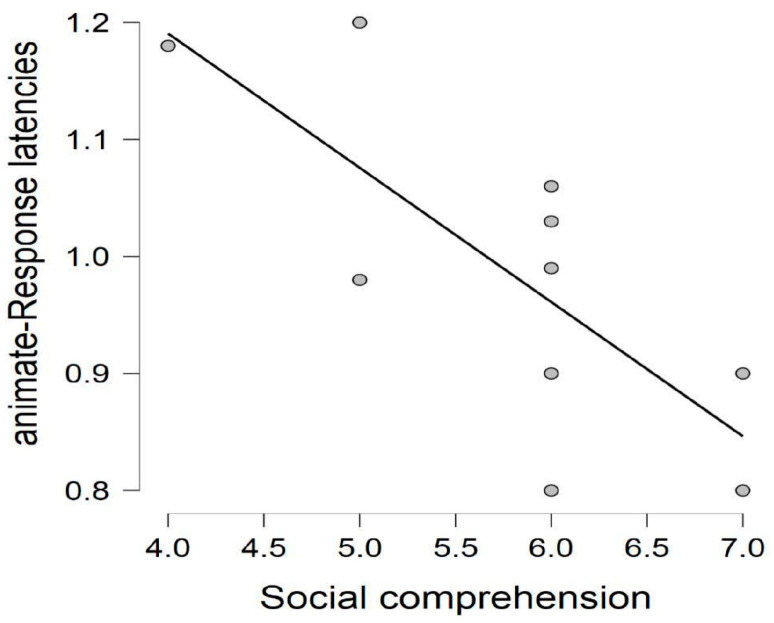
Correlation between the autistic adults’ left frontotemporal slow positive wave amplitudes on the inanimate trials and their scores in the social comprehension task.

**Figure 6 brainsci-13-01656-f006:**
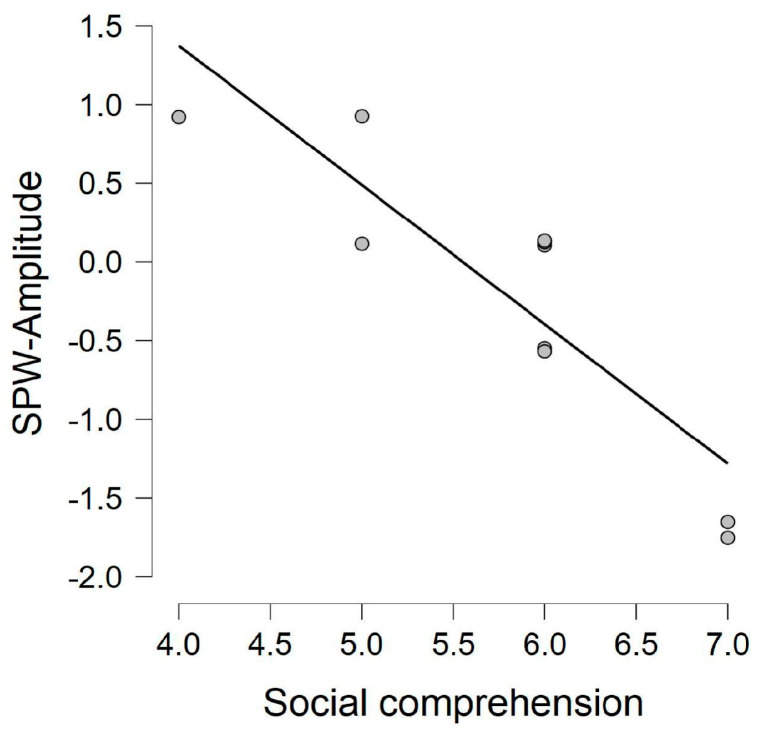
Correlation between the autistic adults’ response latencies on the animate trials of the picture naming task and their scores in the social comprehension task.

**Table 1 brainsci-13-01656-t001:** Participants’ demographic profiles.

	Autistic	Control	*p*-Value
Sex			
Male participants, *n*	2	4	0.820
Mean age, years (*SD*)	21.8 (1.7)	25.4 (3.3)	0.112
Mean full-scale IQ (*SD*)	98.4 (7.9)	102.7 (9.8)	0.220
Autism Diagnostic Interview-Revised			
Mean social interaction (*SD*) (cutoff = 10)	15.5 (2.2)		
Mean communication (*SD*) (cutoff = 8)	12.2 (1.1)		
Mean stereotyped patterns (*SD*) (cutoff = 3)	4.5 (0.6)		

Footnote. *n* = number.

**Table 2 brainsci-13-01656-t002:** Mean accuracy and response latencies (SDs) per group in the animate and inanimate trials of the picture naming task.

	Autistic	Control
Accuracy (%)		
Animate	84.1 (7.5)	91.8 (5.2)
Inanimate	88.6 (6.7)	89.2 (6.5)
Response latencies (in ms)		
Animate	1.09 (0.3)	1.07 (0.3)
Inanimate	1.07 (0.2)	0.98 (0.2)

**Table 3 brainsci-13-01656-t003:** Visualization of statistically significant ERP findings in the picture naming task.

	N100	P100	P200	P300	Slow Negative Wave (SNW)	Slow Positive Wave (SPW)
Left anteriofrontal						
Right anteriofrontal						
Left frontotemporal						
Right frontotemporal						
Left occipital						
Right occipital						
*p* > 0.05						
0.05 ≤ *p* ≤ 0.04						
0.05 ≤ *p* ≤ 0.03						
0.03 ≤ *p* ≤ 0.01						
*p* ≤ 0.01						

## Data Availability

The data presented in this study are available on request from the corresponding author. The data are not publicly available due to the applicable data protection law in Greece (Law 4624/2019).

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
