# Peer review of "Animacy Processing in Autism: Event-Related Potentials Reflect Social Functioning Skills"

_brainsci, 2023, doi:10.3390/brainsci13121656_

Round 1

Reviewer 1 Report

Comments and Suggestions for Authors

The study focuses on investigating the neural mechanisms underlying animacy perception in autistic individuals and its relationship with their social functioning skills. The authors aim to bridge the gap in understanding how individuals with autism process the concept of animacy and how potential deficits might impact their social abilities.

The use of a picture-naming paradigm, coupled with high-density EEG analysis, offers a robust method to assess and map the spatiotemporal dynamics of animacy processing. Matching participants for age and IQ between the autistic and control groups strengthens the study's comparative validity.

This study provides valuable insights into the neural underpinnings of animacy processing in autism and its association with social difficulties. However, further research is necessary to solidify the causal relationship between animacy deficits and social functioning skills. 

I have only two minor comments about the article:

1. Addressing the limitations in sample size and diversity could strengthen the generalizability of the findings.

2. It seems that the images are stretched and their proportions are distorted. The authors should check the resolution and placement of the images within the text of the article.

Author Response

The study focuses on investigating the neural mechanisms underlying animacy perception in autistic individuals and its relationship with their social functioning skills. The authors aim to bridge the gap in understanding how individuals with autism process the concept of animacy and how potential deficits might impact their social abilities.

The use of a picture-naming paradigm, coupled with high-density EEG analysis, offers a robust method to assess and map the spatiotemporal dynamics of animacy processing. Matching participants for age and IQ between the autistic and control groups strengthens the study's comparative validity.

This study provides valuable insights into the neural underpinnings of animacy processing in autism and its association with social difficulties. However, further research is necessary to solidify the causal relationship between animacy deficits and social functioning skills.

Our reply: We would like to thank the reviewer for their encouraging comments. Minor comments have been fully addressed in the revised manuscript.

I have only two minor comments about the article:

  1. Addressing the limitations in sample size and diversity could strengthen the generalizability of the findings.

Our reply: The reviewer’s concern has been acknowledged and currently appears as the first limitation of the study. The following text has been added in the limitations of the study:

First, since the current study included a small sample size of participants, picture naming tasks or paradigms manipulating animacy should be replicated in future studies with larger numbers of participants to ensure that the results are robust. Based on the well-acknowledged heterogeneity in autism and the large individual differences noticed among autistic individuals across studies [58,59], improved sample sizes are required to enable broader conclusions”.

References

Lord, C. Recognising the heterogeneity of autism. Lancet Psychiat. 2019, 6, 551–552.  

Mottron, L.; Bzdok, D. Autism spectrum heterogeneity: fact or artifact? Mol Psychiatry. 2020, 25, 3178-3185.

  1. It seems that the images are stretched and their proportions are distorted. The authors should check the resolution and placement of the images within the text of the article.

Our reply: We apologize for the lack of uniformity across the figures of the manuscript. The figures that represented the ERP waveforms have been replaced by Figure 4.

Reviewer 2 Report

Comments and Suggestions for Authors

The manuscript titled “Animacy processing in autism: Event related potentials reflect: social functioning skills” provided an insight about the correlation between animacy and social score in non-IQ deficit autism group. Although it showed some interesting results, what the exactly biological meaning of this result for the autism research is not clear. In addition, some data presentations need to be very carefully addressed. Here are my comments:

1. The authors mentioned number of participants in each group. Did the data from all subjects passed the quality control? What’s the n number in each graph?

2. It’s hard to define a clear ASD subtype to get a general idea what kind of ASD type was under investigation with only IQ, age and gender information. What’s the other ASD related score from clinical diagnosis?

3. The introduction was written like a review paper, which is very hard to follow the main points authors want to present. It’s better to concentrate the ideas which were related to the research focus.

4. The authors need to very clearly showed the experiments design with graph and list the pictures used in the task. The cited reference was not helpful to understand the design.

5. The data from ERP was analyzed by two-way Anova based on the description in the methods part. However, value from each subject is essential to showed the mean and individual difference. Only a window with p-value was not clear enough.

6. The most concern is the inconsistency analysis across either left/right frontotemporal electrode cluster. The peak or window to be analyzed were different, although the authors mentioned only showed difference between group. The similar analysis was found in occipital electrode as well.  It’s hard to interpreter these conclusions. A predesigned or predicted matrix should be carefully though before the data analysis. Otherwise, it will be very difficult for other researches to reproduce it in the further.

7. The author only showed from “frontotemporal Slow Positive Wave amplitude on inanimate trials and their scores in the social comprehension task”. Why not all the other finding from the ERP?

8. For the naming accuracy in the behavior result, the result plot is very essential. It’s confusing based on author’s description.

Round 2

Reviewer 2 Report

Comments and Suggestions for Authors

The manuscript is significantly improved. The quality is well fit Brain Sciences.